# Bioconversion of Corn Crop Residues: Lactic Acid Production through Simultaneous Saccharification and Fermentation

**Alonso Malacara-Becerra** [1,2], **Elda M. Melchor-Martínez** [1,3,*], **Juan Eduardo Sosa-Hernández** [1,3],
**L. María Riquelme-Jiménez** [1,2], **Seyed Soheil Mansouri** [2], **Hafiz M. N. Iqbal** [1,3] and **Roberto Parra-Saldívar** [1,3,*]

1   Tecnologico de Monterrey, School of Engineering and Science, Monterrey 64849, Mexico
2   Process and Systems Engineering Center (PROSYS), Department of Chemical and Biochemical Engineering, Technical University of Denmark, 2800 Kgs Lyngby, Denmark
3   Tecnologico de Monterrey, Institute of Advanced Materials for Sustainable Manufacturing, Monterrey 64849, Mexico
*   Correspondence: elda.melchor@tec.mx (E.M.M.-M.); r.parra@tec.mx (R.P.-S.)

**Abstract:** Lactic acid (LA) is a chemical building block with wide applications in the food, cosmetics, and chemical industries. Its polymer polylactic acid further increases this range of applications as a green and biocompatible alternative to petrol-based plastics. Corn is the fourth largest crop in the world, and its residues represent a potentially renewable feedstock for industrial lactic acid production through simultaneous saccharification and fermentation (SSF). The main goal of this work is to summarize and compare the pretreatment methods, enzymatic formulations and microbial strains that have been combined in a SSF setup for bioconversion of corn crop residues into LA. Additionally, the main concerns of scaling-up and the innovation readiness level towards commercial implementation of this technology are also discussed. The analysis on commercial implementation renders the current state of SSF technology unsustainable, mainly due to high wastewater generation and saccharification costs. Nonetheless, there are promising strategies that are being tested and are focused on addressing these issues. The present work proves that the study and optimization of SSF as a biorefinery framework represents a step towards the adoption of potentially sustainable waste management practices.

**Keywords:** second-generation lactic acid; corn stover; corncob; simultaneous saccharification and fermentation; lignocellulose revalorization

## 1. Introduction

Agricultural production is one of the main economic activities in the world, accounting for 5% of the worldwide gross domestic product (GDP) in 2020, and over 25% in most developing countries [1] This makes lignocellulosic biomass (LB) derived from crops an important waste stream in most countries. A small portion of the waste is used as animal feedstock or compost, while most of it is disposed by landfilling or burning, contributing to greenhouse gases (GHGs) [2–4]. Additionally, improper disposal and accumulation leads to the contamination of water effluents by acidification and rise of chemical oxygen demand [5]. Thus, sustainable management of agricultural waste streams represents a priority in the reduction of the environmental footprint of human-associated activities.

The circular economy (CE) concept emerged in the 1970s as a framework based on three principles: eliminate waste and pollution, circulate products and materials (at their highest value) and regenerate nature [6]. In this sense, biorefinery of LB is an alternative that is in line with the three CE principles. From the perspective of biorefinery, agro-industrial biomass is not considered waste and its polluting disposal practices are replaced by the transformation into value-added bioproducts (Figure 1).

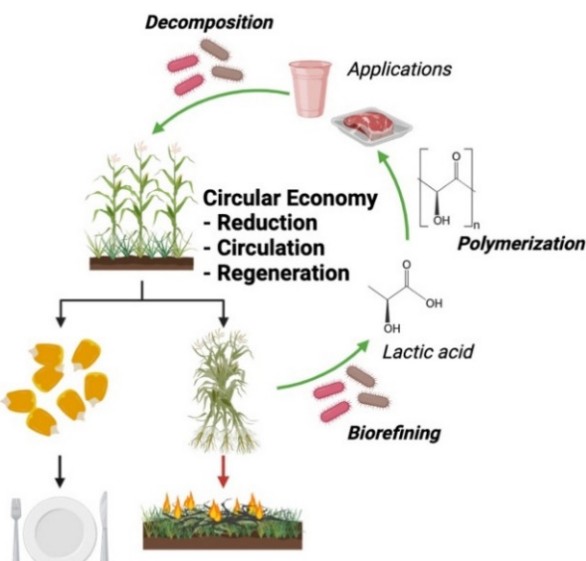

**Figure 1.** Example of biorefining of agro-industrial biomass as a framework for circular economy.

Biorefinery is based on the fermentation of the carbohydrates present in biomass, such as glucose and xylose, to produce a bioproduct of higher value. Lactic acid (LA) is a product of the central carbon metabolism of different bacteria and fungi species [7,8]. LA is considered a commodity chemical, with a wide range of industrial applications that are determined by: its acidic character in aqueous medium, its bifunctional reactivity linked to the presence of a hydroxyl and a carboxyl group, and its optical activity [9].

L-LA is generally recognized as safe (GRAS) by the Food and Drug Administration (FDA) and used in the food and beverage industry [10]. Additional applications in the cosmetics and hygiene industry have been documented [9–12]. The leading demand for LA is in the production of polymers as precursor of polylactic acid (PLA), a biodegradable alternative to petroleum-based plastics [13]. The degradation activity of L-lactate dehydrogenase inside the human body allows for biomedical [7], pharmaceutical applications [14,15] and drug delivery matrices [16].

Mass production of LA uses fermentation of corn, beet and other first-generation (1G) feedstocks for the simplicity of the upstream operations. On the other hand, second- (2G) and third-generation (3G) feedstocks, such as LB and algae biomass, respectively, are cheaper substrate options that do not compete with food, and are instead often considered waste [17].

Different approaches for 2G-LA production have emerged (Figure 2). The first one is called separate hydrolysis and fermentation (SHF) and involves enzymatic hydrolysis of the polysaccharides and subsequent fermentation of the released sugars. In contrast, simultaneous saccharification and fermentation (SSF) and consolidated bioprocess (CBP) are one-pot methods, where enzymatic hydrolysis and microbial fermentation occur in parallel [18,19]. The main advantage of combining both operations is that hydrolytic activity is increased due to reduction in feedback inhibition (main limitation of SHF) by immediate product consumption [20].

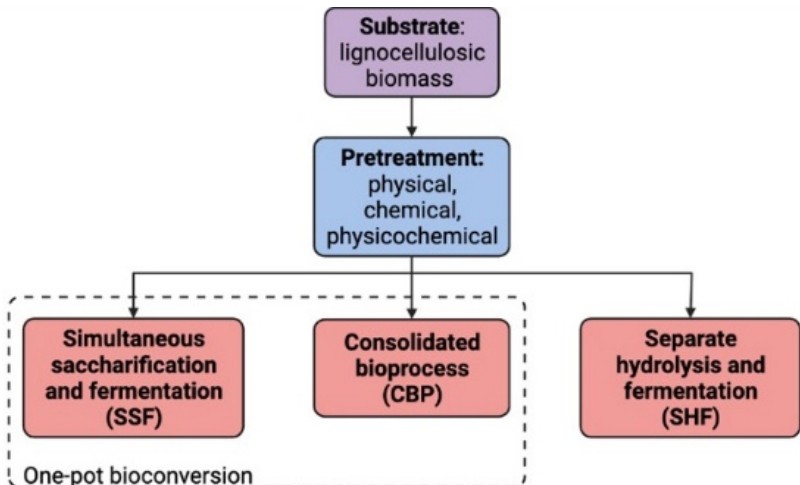

**Figure 2.** Lignocellulose revalorization methods. The lignocellulosic feedstock enters as a substrate, goes through a pretreatment process and either the SHF, CBP or SSF may be applied for the obtention of LA.

In SSF, fungi-derived free cellulases are used for off-site saccharification of the complex carbohydrate matrix into a fermentable carbon source [20]. Meanwhile, CBP consists of the selection of microorganisms that do not only ferment monosaccharides to the desired product, but also produce the enzymes needed for the saccharification [19].

Apart from increased hydrolytic activity, SSF provides operational advantages (discussed in more detail in Section 2.2) that in practice result in higher yields, LA titers and productivity when compared to SHF bioconversion [21]. This is also reflected in a higher minimum product selling price (MPSP) of LA manufactured by SHF [22] compared to the SSF scheme [23]. The main limitation of SSF is that a compromise must be often made between hydrolytic and fermentation rate because of the difference in reaction temperatures [24]. Nonetheless, experimental results favor SSF most of the time. This has caused an increase in the number of studies published in recent years, establishing a trend in SSF and its variations.

According to the Food and Agriculture Organization (FAO), maize is the fourth largest food crop by weight in the world [25]. The residues from harvesting, such as corn stover, are an interesting feedstock to produce bio-based chemicals due to their worldwide availability and relatively high amounts of carbohydrates. Several experiments are being carried out to harness the carbohydrates present in this LB [26].

This paper is a revision of the state-of-the-art LA production through SSF of agricultural residues of corn. The advantages and disadvantages of different pretreatment methods have been evaluated. The main saccharification enzymes, microorganisms employed and challenges faced by LB revalorization and promising technologies are discussed. Lastly, the maturity level of the technology is defined using a technology readiness level-based methodology specific to chemical manufacturing.

## 2. Bioconversion Process

Bioconversion is the use of the biochemical energy stored in waste biomass to produce chemicals of higher value through fermentation. Different frameworks of bioconversion surged as alternatives for biomass disposal, which depend on the nature of the residue. Bioconversion of LB includes three key operations: pretreatment, saccharification and fermentation. When each operation is performed independently, the process is called SHF. On the other hand, the saccharification and fermentation operations can be coupled into a single vessel operation, called SSF (Figure 2). Each of the operations and their research as a part of SSF on corn crop residues will be discussed in more detail in the present section.

## 2.1. Pretreatment

Valorization of crop residues requires a pretreatment step to increase the permeability of the matrix for the subsequent enzymatic hydrolysis. Chemical factors of LB, namely the content of cellulose, hemicellulose, lignin, and the interaction among these biopolymers, create a heterogeneous matrix that is naturally recalcitrant to enzymatic conversion. This is further influenced by physical factors of LB, such as crystallinity, particle size, specific surface area and pore size [27]. Both polymers, cellulose and hemicellulose, are potential biofactory feedstocks, which leaves lignin and its interlinkages with both polysaccharides as the main targets of the pretreatment.

In addition to enabling maximum access of hydrolytic enzymes to their substrate, the following set of considerations (adapted from the one originally proposed in [28]) should serve as a guideline for the selection of an appropriate pretreatment method: (a) it should minimize energy demands, time and unit operations in the overall process, (b) hemicellulose fraction must be preserved if subsequent stages allow its bioconversion, (c) degradation byproducts must be minimized, (d) the pretreatment catalyst should be low-cost itself and/or its recycling should be inexpensive, and (e) its environmental impact and water usage should be kept to a minimum. The approaches for LB pretreatment fit within one of the following categories: physical, chemical, or physicochemical operations; being the combination of two or more methods as a recurring strategy.

### 2.1.1. Physical Pretreatment

Methods for physical pretreatment of LB usually generate fewer waste residues and fermentation inhibitors compared to chemical and physicochemical pretreatments. Physical methods can be further categorized into size reduction operations, thermal (pyrolysis) and field-assisted treatments [28]. Detailed information on the latter two can be found in [29]. The reviewed SSF experiments were limited to size reduction operations such as ball milling and extrusion. Except for the experiments in [30], size reduction was always followed by a chemical or physicochemical pretreatment.

Size reduction increases the surface area accessible to hydrolytic enzymes, and it can also decrease crystallinity of lignocellulose. Although it hardly produces any chemical changes on its complex polymeric matrix [28], size reduction is still necessary to enhance digestibility and fluidity of LB [31]. For SSF, corn-derived feedstocks are milled between 0.075 [32] and 10 mm [23,33–35]. Finer milling is often associated to improved cellulose conversion efficiency. However, this translates to increasing operational costs. Moreover, hydrolysis yield remains approximately constant in saccharification of corn stover milled to fractions between 0.21 and 1.42 mm [36]. This indicates that particle size is a minor conversion factor in SSF, and its optimization should be limited to the selection of low-energy-demanding milling methods.

### 2.1.2. Chemical Pretreatment

Chemical pretreatment of LB can include dilute acid, mild alkali, ozonolysis, and deep eutectic solvents, among others. Discussion in this section is limited to acidic and alkali methods, because only these have been used as part of the bioconversion framework under investigation.

Acid pretreatment at low temperatures (<100 °C) requires concentrated solutions (30–70%). On the other hand, dilute acids (0.1–10%) can also be used under higher temperatures (100–250 °C). Both sets of conditions are harsh to the substrate and might cause the generation of inhibitory compounds, such as phenolic compounds, furfurals and aldehydes [37]. These substances are responsible for large amounts of wastewater generation. Dilute acid pretreatment is preferred because of its diminished toxicity and corrosion risks, as well as its lower maintenance costs. Acid pretreatment is also particularly effective against hemicellulose, which has a positive impact on the recovery of the cellulosic fraction.

Contrary to dilute acid, alkali pretreatment under mild conditions has been demonstrated to maintain most of the hemicellulose fraction of LB, while at the same time having a

significant impact on the solubilization of lignin. Xylose (hemicellulose) utilization must be pondered to attain maximum substrate utilization. For example, 5% (*w/w*) NaOH-pretreated corn stover showed an increase of 61.66% in cellulose content, 66.97% decrease in lignin content, and 7.67% decrease in hemicellulose composition, with respect to the raw feedstock. After a fed-batch SSF, up to 0.77 g of LA/g of stover were obtained, demonstrating the potential results when a xylose fermenting microorganism is used (Table 1) [38].

Alkali reagents act against the uronic acid linkages between hemicellulose and lignin [39]. Moreover, glycosides and other intermolecular ester bonds are also affected. These solutions also promote cellulose swelling and decrystallization, which increases internal surface area [28]. Dilute NaOH is the most employed alkali catalyst in pretreatment of corn crop residues, because it has been proven efficient for pretreatment of LB with less than 26% lignin [28].

By far, dilute acid and mild alkali methods are the most popular pretreatments for corn crop residues bioconversion into LA. Their implementation requires low investment, easy operation, and they effectively reduce the recalcitrance of corn crop residues. Unfortunately, scaling-up remains environmentally and industrially challenging, considering the large amounts of water needed to remove fermentation inhibitors from the pretreated feedstock. The main approaches to address scalability limitations of chemical pretreatment are discussed in Section 3.1.1.

2.1.3. Physicochemical Pretreatment

Physicochemical methods include: ammonia-based pretreatments, autohydrolysis (also referred to as steam explosion), liquid hot water, oxidative operations, etc. These methods are less frequently used for the pretreatment of corn crop residues. Soaking aqueous ammonia and autohydrolysis pretreatments have been followed by SSF of corn stover [40] and corncob [41], respectively.

Ammonia-based pretreatments are differentiated according to the temperature and pressure at which they take place. Soaking aqueous ammonia (SAA) occurs at temperatures between 30 and 70 °C and at atmospheric pressure. Ammonia induces selective delignification, resulting in minor hemicellulose and cellulose degradation [42]. Ammonia also acts by swelling the lignocellulosic matrix. An environmental advantage of this method is that aqueous ammonia can be reused for pretreatment of subsequent batches [43]. On the other hand, the nature of this catalyst requires a washing step of the biomass before its saccharification.

A protocol of corn stover soaking in aqueous ammonia, where long treatment time and a high temperature were selected (90 °C for 24 h), showed a significant decrease in lignin content (from 17.2 to 7.7%). In absolute terms, most of the glucan and xylan fractions were maintained, whereas in relative terms these fractions went from 36.8 to 54.4% and from 21.7 to 24.9%, respectively [40].

Autohydrolysis pretreatment is carried out with saturated steam, at pressures between 0.69 and 4.83 MPa, and temperatures of 160 and up to 260 °C. These extreme conditions allow water to permeate into the lignocellulosic structure. This way, a sudden drop of pressure induces the absorbed water to swiftly escape, causing an explosion that affects the lignocellulosic fibers at a structural level. The action mechanism allows short treatment times, in the order of seconds to minutes. In addition to said mechanical effect, acetyl groups from side chains of hemicellulose generate acetic acid. Acetic acid then acts on the glycosidic bonds of hemicellulose, to release glucose and xylose [29].

This pretreatment method requires fewer chemicals (acidic catalysts might be added), and its short treatment times are reflected in low energy usage. Corncobs were pretreated by autohydrolysis in a Parr reactor at 202 °C, to obtain a hemicellulose-free substrate for SSF. The high temperature resulted in a decrease in hemicellulose content, from 39.0 to 10.4%. The relative decrease of hemicellulose caused an increase in cellulose and lignin, from 39 to 59.1%, and from 14.4 to 22.6%, respectively. This means that the lignin was not completely digested. In the same experiment, 8 g of water/g of dried corncob were used for the autohydrolysis operation [41]. Therefore, another main disadvantage of this technique is high water usage. This is further impacted because, as with the rest of the

chemical and physicochemical methods, fermentation inhibitors might be generated at high temperatures, which calls for a washing step.

The pretreatment step represents an area of opportunity in the bioconversion of LB. As explored in this section, this operation is associated with the generation of harmful byproducts and high demand of resources. Additionally, the pretreatment of corn stover requires an investment twice the value of what would be required for the bioconversion of third-generation feedstock (seaweed). First-generation (glucose) substrates do not require a pretreatment step [44]. Therefore, the development of a framework capable of reducing pretreatment-associated costs would have a direct impact in increasing the economic competitiveness of 2G-LA.

*2.2. Simultaneous Saccharification and Fermentation*

The main advantages of SSF for industrial production of LA include [45]:

1.    It avoids the need to physically separate hydrolysate from biomass, an operation that would inevitably lead to sugar loss;
2.    In comparison to SHF, larger amounts of pretreated LB can be fed into the bioreactor at once;
3.    Performing two steps simultaneously in the same reactor has a direct impact on reducing overall process duration;
4.    Investment costs are reduced because fewer reaction vessels are needed;

On the other hand, one of the main design challenges is finding an optimal combination of enzymes and microorganisms. Ideally, the microorganisms and the hydrolytic enzymes need to fit within each other's temperature and pH range. Otherwise, a trade-off between enzymatic yield and fermentation yield would be observed [24]. Most saccharification enzymes show their highest activity at temperatures between 45 and 50 °C [46]. Additionally, materials and equipment are the same in SHF and SSF. Therefore, favoring SSF over SHF for LA production should be the case as long as a thermotolerant LAB strain is available.

The selected microorganism must produce high-titer and optically pure LA, with low formation of byproducts. The fermentation operation requires a minimum LA titer of 100 g/L and over 99% optical purity for an industrial-scale process to be economically feasible. This value enables for cost-efficient downstream operations [23].

2.2.1. Enzymatic Hydrolysis

Once the pretreatment step is over, polysaccharides from LB are more readily accessible for hydrolysis. This operation consists of the monomerization of cellulose and hemicellulose into fermentable sugars by hydrolytic enzymes. Saccharification of LB is performed using different free enzymes, which focus on hydrolyzing different linkages that comprise the polymeric structure of cellulose and hemicellulose. Enzymatic mixtures are thus needed to enable maximum and efficient co-hydrolysis of the polysaccharide matrix.

The main group of enzymes used for complete saccharification of cellulose are cellulases. This term includes three different types of enzymes, which are endoglucanase (E.C. 3.2.1.4), exoglucanase (also referred to as cellobiohydrolase) (E.C. 3.2.1.91) and β-glucosidase (also referred to as cellobiase) (E.C. 3.2.1.21) [47]. Endoglucanase acts on the β-1,4 D-glycosidic bonds in cellulose. Exoglucanase also hydrolyzes β-1,4 D-glycosidic bonds but its activity is limited to cleave off cellobiose units from the ends of the polymeric chain. Lastly, β-glucosidase transforms each cellobiose molecule into two glucose molecules [48]. Additionally, β-glucosidase enhances the activity of the other two enzymes, as they are inhibited by cellobiose [49].

Hemicellulases are often regarded as accessory enzymes for saccharification, but have been proven essential to assist on the retrieval of fermentable sugars. They are needed to facilitate cellulase access to cellulose fibers, by breaking the hemicellulose barrier. Moreover, their hydrolysate from corn crop residues has the potential to be converted into LA, given that a pentose-fermenting microorganism is introduced. When this is the case, the process is often called simultaneous saccharification and co-fermentation (SSCF). Xylose is the main

monosaccharide in the hemicellulose fraction of corn crop residues [50], and it is present in its polymeric form as xylan. Therefore, the appropriate hemicellulases for this kind of feedstock are endoxylanase and β-xylosidases. The first one hydrolyzes the xylan backbone of hemicellulose into short oligosaccharides, while the latter one catalyzes the monomerization of these oligosaccharides into xylose [51]. Thus, effective lignocellulose deconstruction is a product of the synergistic work of a broad spectrum of hydrolytic enzymes.

Follow-up studies should assess different enzymatic formulations, based on specific feedstocks and fermentation configurations. Since the saccharification rate has been demonstrated to be a limiting step in SSF, taking the time to find an optimal composition could lead to an efficient and cost-effective bioconversion [52]. One example of such experiments was conducted in [53], where hemicellulose-free corncob residue (from acid pretreatment) was hydrolyzed with different formulations of cellulases (which included endo- and exogluganase) and β-glucosidase. Under identical SSF conditions, the highest LA titer was achieved at loadings of 2:1, in terms of U/g cellulose. Out of the reviewed works, this was the only one where the proportion of the enzymes was a factor under investigation.

Most of the studies on SSF of herbaceous residues utilize commercial mixtures of enzymes. Manufacturers rarely make information about the enzymatic composition of their products available to the customer. This limits the factors related to saccharification that can be tested and therefore, hydrolysis optimization strategies are also narrowed. Furthermore, Cellic® CTec2 (Novozymes, Bagsværd, Denmark) was used as an enzymatic cocktail in 50% of the experiments selected for this review. According to the supplier (Novozymes, Application sheet Luna No. 2010-01668-01), the optimal temperature and pH of this product are 45–50 °C and 5.0–5.5, respectively.

Enzymatic cocktails available in the market are composed of cellulases from highly productive filamentous fungi such as *Aspergillus niger*, *Trichoderma longibrachiatum* and *Trichoderma reesei* [54]. The proteome analysis of these kind of organisms has allowed for the identification and improvement of saccharification cocktails. For instance, over 30 LB degrading enzymes were found in the fermentation broth of *T. reesei* TUT C-30, when pretreated corn stover was used as fermentation feedstock [55]. Out of these, six enzymes are considered essential for deconstruction of pretreated corn stover: cellobiohydrolase I and II (CBHI and CBHII), endoglucanase I (EGI), β-glucosidases (βG), endo-β-1,4-xylanases (EX) and β-xylosidases (βX) [55,56]. In addition to these, other lignocellulosic and accessory enzymes might be present in the benchmark formulations. Among them are endoglucanase III, endoglucanase IV, endoglucanase VII, exoglucanase II and chinitase [57]. The formulation and sources of the commercial cocktails are not available to consult. However, Cellic® CTec2 (Novozymes, Bagsværd, Denmark) has demonstrated strong exoglucanase, β-glucosidase and xylanase activities at 50 °C and pH 4.8 [58].

It is important to note that out of the reviewed experiments that used Cellic® CTec2, only [59] was performed at 50 °C, and the experiment in [60] was conducted within the optimal pH range. Thus, most experiments prioritize fermentation over hydrolysis conditions. In some cases, such experimental decisions might be empirical, where residual carbohydrates might be relatively present when fermentation is over. No cases have been reported where pH and temperature are experimental factors with varying levels in a SSF experiment. Instead, using corn-derived feedstock, some researchers have opted to include a 6 h saccharification step at 50 °C prior to SSF at 43 °C [23,33–35,61]. This strategy seems to be associated with high LA titers and yields.

With the available reports, it is difficult to define an optimal enzymatic cocktail for SSF of corn crop residues. The reviewed experiments focused the attention on the end product variables rather than intermediate operations. However, Cellic® CTec2 (Novozymes, Bagsværd, Denmark) has been showed to outperform other formulations in the hydrolysis of multiple substrates [54], including corn crop residues [62].

### 2.2.2. Fermentative Microorganisms

To maximize fermentation yield and LA titer, a variety of wild-type and genetically engineered microorganisms have been studied for bioconversion of corn crop residues. The main interest lies in LA bacteria (LAB). LAB ferment glucose to LA through glycolysis (or the Embden–Meyerhof–Parnas pathway; EMPP), and some pentoses through the pentose phosphate pathway (PPP); additionally, the presence of phosphoketolase enzymes might redirect the latter to produce acetate and ethanol (Figure 3). Depending on their metabolic profile, they are classified as homofermentative (hexoses uptake and LA production), facultatively heterofermentative (pentoses/hexoses uptake and production of LA and other products) and heterofermentative (pentoses/hexoses uptake and production of LA, side products and $CO_2$) [63]. Mixed bacterial and mixed fungal fermentations have also been explored as a means to maximize saccharification and fermentation yields.

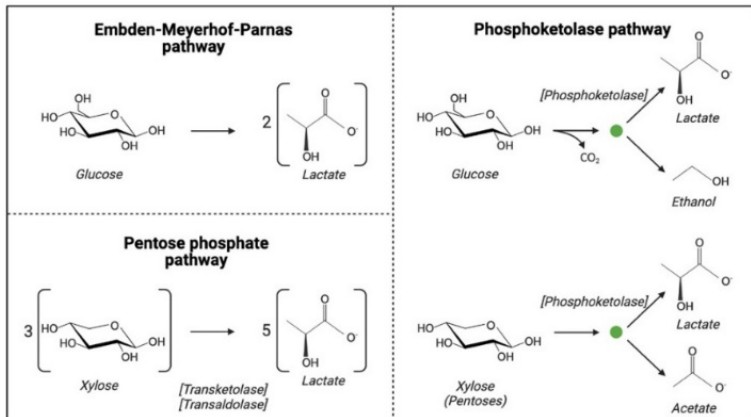

**Figure 3.** LAB shows production of LA through three main pathways, which might share some molecular intermediates. The green dot (·) represents the divergence from the PPP. Only the overall substrates and products of each pathway are shown. The potential metabolic input and output is commonly used as a classification criterion in LAB strains.

Wild-Type Microorganisms

*Lactobacillus pentosus* is a mesophilic species, with a facultative heterofermentative metabolism [64]. In the experiments in [40], it was reported that the glucose and arabinose uptake rate remained high during batch SSF experiments using *L. pentosus* ATCC 8041. However, significant xylose consumption occurred only after glucose depletion, which was attributed to diauxic growth. At the same time, LA production slowed down, while acetic acid productivity increased. In the fed-batch culture, LA accumulated rapidly before 72 h, but a sudden drop in productivity was observed before a concentration of 60 g/L was reached. The researchers concluded that this was due to the inhibitory effect of the lactate ions on *L. pentosus*.

In contrast, *L. pentosus* FL0421 was tested in a fed-batch SSF experiment, and simultaneous consumption of glucose and xylose was observed [65]. In the same study, the authors presumed that the concentration of xylose dissolved in the medium had an effect on the preferred metabolism pathway for this pentose. It was noted that at a high concentration of xylose, the productivity of acetic acid was significantly higher compared to that of a low xylose concentration. A final concentration of 34.27 g/L of acetic acid was produced. They proposed a low-rate initial saccharification stage to overcome this phenomenon.

Other LAB have also been tested for their attractive fermentation features. *Pediococcus acidilactici* had already been used for production of bacteriocin [66], but it was not until recently that the species attracted attention for its high-titer LA production. The first reported SSF of pretreated corn residue, using a *P. acidilactici* strain, was performed in [24]. The fermentation of pretreated corn stover by *P. acidilactici* DQ2, resulted in a downstream-cost-efficient LA concentration of 101.9 g/L. This isolated strain also showed optimal LA

productivity at 48 °C and low metabolic inhibition by compounds generated in acidic pretreatment of LB. However, optical purity of the product was poor (L-LA at 63.4%) and xylose utilization was observed only at non-optimal growth conditions. These issues were addressed through genetic engineering in later experiments that will be discussed in the following section. Similar LA titer results were obtained using *P. acidilactici* PA204 (104.11 g/L) [38]. In this case, LA productivity seemed to be affected by alkali pretreatment inhibitors of corn stover. This strain was able to convert xylose into LA, but at a much slower rate compared to glucose conversion. Even so, more than half of the fed xylose fraction remained at the end of the experiment. It is worth noting that acetic acid formation by *P. acidilactici* PA204 was minimal.

The most studied wild-type bacteria for LA production is *Bacillus coagulans*. Different strains have shown optimal growth and LA production at 50 °C [53,59,67,68]). This physiological characteristic is compatible with the temperature needed for high rate saccharification by cellulase activity and it also enables the maintenance of axenic cultures under non-sterile conditions. *B. coagulans* is also associated with: xylose uptake [32], low carbon catabolite repression [59] and low byproduct formation [53]. It has also been proven to produce enantiomeric pure L-LA (>99%), and to be resistant to growth inhibitors that are generated by the acidic pretreatment of corn fiber [67]; whereas phenolic compounds formed during alkali pretreatment seem to hinder the fermentation [32].

Less studied microorganisms include *Lactobacillus delbrueckii,* which was selected based on its ability to produce optically pure D-LA (99.9%) under anaerobic conditions. However, it was not able to utilize xylose, and its growth temperature and pH did not match those of the selected saccharification enzymes [69]. This is also the case with *Lactobacillus rhamnosus*, which instead exhibited a metabolism of cellobiose, avoiding the need to include β-glucosidase in the enzymatic cocktail while at the same time reaching a cellulose conversion of up to 97.5% [41].

Co-culture methods have been suggested as a means to widen the range of fermentable sugars from cellulose and hemicellulose hydrolysate. The simultaneous inoculation of *Lactobacillus rhamnosus* and *Lactobacillus brevis* showed an increase of 18.6 and 29.6% in LA yield (with respect to their monocultures) from a batch fermentation of pretreated corn stover. The selection was made on the argument that *L. brevis* is able to utilize glucose and xylose simultaneously, while *L. rhamnosus* competed for glucose intake and avoided high byproduct formation [70]. Under the same logic, a mixed culture of *L. brevis* ATCC 367 and *Lactobacillus plantarum* 21028 was tested [71]. In this case, *L. plantarum* was inoculated 24 h prior to *L. brevis*. This approach allowed maximum glucose conversion to LA, and utilization of xylose. In both experiments, the SSCF temperature was kept at 37 °C, which reduced hydrolytic enzymes activity by approximately 40% when compared to hydrolysis under optimal temperature (45–50 °C).

*Acremonium thermophilus* is a thermophilic fungus that can grow at extreme temperatures and has not been as studied as much as other thermophilic species but is recognized as a potential source of enzymes with scientific and commercial benefits [72]. In particular, the *A. thermophilus* ATCC 24622 strain produces cellulose by hydrolysis and has proved its efficiency for LA production [73]. In [30] culture samples of the fungus were used, thus obtaining the highest yields of L-LA from the untreated substrate. LA was produced using *Acremonium* cellulase and its enhanced production was achieved by SSF with a mixed culture of *A. thermophilus* and *Rhizopus* sp., without the addition of cellulase preparation.

*Rhizopus oryzae* NLX-M-1 is a filamentous fungus that has been used to ferment pretreated corncobs with low hemicellulose content [74]. Although the fungus produces optically-pure L-LA from glucose as well as from xylose (showing carbon catabolite repression), it also exhibits important drawbacks with respect to its bacterial counterparts. First, *R. oryzae* is an aerobic organism, which inevitably increases manufacturing costs and second, its growth temperature is around 30 °C, a value at which cellulase activity is hindered. Finally, a considerable amount of other fermentation products, such as ethanol could be generated (up to 0.24 g of ethanol for each gram of LA in a high-solids loading experiment).

**Table 1.** Methodology and results from SSF experiments using wild-type microorganisms.

| Microorganism | Feedstock | Pretreatment [a] | Saccharification | Fermentation Conditions | Lactic Acid | | | | Xylose Utilization | Reference |
|---|---|---|---|---|---|---|---|---|---|---|
| | | | | | Yield | Titer (g/L) | Productivity (g/L·h) | Optical Purity (%) | | |
| *Bacillus coagulans* H-1 | Corncob residue | Dilute acid | Cellulase cocktail (10 U/g cellulose) supplemented with β-glucosidases (15 U/g cellulose) | Batch culture at 50 °C, pH 6.0 and 10% (*w/v*) solids loading | 0.85 [c] | 68.0 | 1.89 | 100 [g] | No | [53] |
| *Bacillus coagulans* L-LA 1507 | Corn stover | NaOH pretreatment, at 118 °C for 1 h. | Commercial cocktail of cellulases from KDN Group | Sixth subsequent fed-batch cultures with recycled streams, at 50 °C and pH 6.2 | 0.396 [b, d] | 93.42 [e] | 2.14 [d] | N. R. | Yes | [68] |
| *Bacillus coagulans* MXL-9 | Corn fiber | Dilute acid (H$_2$SO$_4$) pretreatment, at 121°C for 1 h. | Mixture of Celluclast® 1.5 L and Novozyme 188. Saccharification 18 h prior to inoculation | Batch culture at 50°C, pH 6 and 10% (*w/v*) solids loading | N. R. | 45.6 | 0.21 | >99 [g] | Yes | [67] |
| *Bacillus coagulans* LA204 | Corncob | NaOH pretreatment, at 75 °C for 3 h NH$_3$ for 1 day, and H$_2$O$_2$ for 7 days, without washing step | Cellic® CTec2 at 30 FPU/g DM | Fed-batch culture at 50 °C, pH 6 and 16% (*w/w*) solids loading | 0.77 [b]<br>0.74 [b] | 122.99<br>118.60 | 1.37<br>1.32 | 98 [g]<br>98 [g] | Yes | [32] |
| *Bacillus coagulans* LA204 | Corn stover | NaOH pretreatment, at 75 °C for 3 h | Cellic® CTec2 at 30 FPU/g DM | Fed-batch culture at 50 °C, pH 6 and 14.4% (*w/w*) solids loading | 0.68 [b] | 97.59 | 1.63 | 94.5 [f, g] | Yes | [59] |
| *Bacillus coagulans* LA204 | Corncob | NH$_3$ for 2 days, and H$_2$O$_2$ for 7 days and biodetoxification | Enzymes isolated from a *Trichoderma viride* culture at 2.0 FPU/g DM | Fed-batch culture at 8% (*w/w*) solids loading | 0.54 [b] | 64.95 | 0.57 | N. R. | Yes | [57] |

**Table 1.** *Cont.*

| Microorganism | Feedstock | Pretreatment [a] | Saccharification | Fermentation Conditions | Lactic Acid | | | | Xylose Utilization | Reference |
|---|---|---|---|---|---|---|---|---|---|---|
| | | | | | Yield | Titer (g/L) | Productivity (g/L·h) | Optical Purity (%) | | |
| *Pediococcus acidilactici* DQ2 | Corn stover | Dry dilute sulphuric acid pretreatment and biodetoxification | Accellerase 1000 at 15 FPU/g DM. Saccharification 8 h prior to inoculation | Batch culture at 48 °C, pH 5.5 and 27% (*w/w*) solids loading | 77.2 [d] | 101.9 | 1.06 | 63.4 [g] | Yes (at specific conditions) | [24] |
| *Pediococcus acidilactici* PA204 | Corn stover | NaOH pretreatment, at 75 °C for 3 h | Cellic® CTec2 at 30 FPU/g DM | Fed-batch culture at 37 °C, pH 6.0 and 15% (*w/w*) solids loading | 0.69 [b] | 104.11 | 1.24 | N. R. | Yes | [38] |
| *Lactobacillus pentosus* FL0421 | Corn stover | NaOH pretreatment, at 75 °C for 3 h | Cellic® CTec2 at 30 FPU/g DM | Fed-batch culture at 37 °C, pH 6.0 and 14% (*w/w*) solids loading | 0.66 [b] | 92.30 | 1.92 | 98.1 [g] | Yes | [65] |
| *Lactobacillus pentosus* ATCC 8041 | Corn stover | Aqueous ammonia at 90 °C for 24 h | Spezyme CP cellulase at 5 FPU/g glucan | Fed-batch culture at 37 °C | 65 | 74.8 | 0.7 | N. R. | Yes | [40] |
| *Lactobacillus delbrueckii* ATCC 9649 | Corn stover | NaOH pretreatment, at 121 °C for 30 min. | Cellic® CTec2 at 8 FPU/g DM | Batch culture at 40 °C, pH 5.5, 4% (*w/v*) solids loading | 0.58 [b] | 20.1 | 0.32 | 99.9 [h] | No | [69] |
| *Lactobacillus rhamnosus* CECT-288 | Corncob | Autohydrolysis with water at 202 °C | Celluclast® at 30 FPU/g DM | Batch culture at 45 °C and 16.7% (*w/v*) solids loading | 86.5 [d] | 86.15 | N.R. | N. R. | N. R. | [41] |
| *Lactobacillus rhamnosus* + *Lactobacillus brevis* ATCC367 | Corn stover | NaOH pretreatment, at room temperature for 12 h | Spezyme CP at 25 FPU/g glucan | Batch culture at 37 °C, pH 5 and 3% (*w/v*) solids loading | 0.70 [b] | 20.95 | 0.58 | N.D. | Yes (by *L. brevis*) | [70] |
| *Lactobacillus plantarum* ATCC 21028 + *Lactobacillus brevis* ATCC 367 | Corn stover | NaOH pretreatment at 121 °C for 30 min | Cellic ® CTec2 at 8 FPU/g DM | Batch culture at 37 °C, pH 6.0 and 4% (*w/v*) solids loading | 0.78 [b] | 31.2 | 0.43 | 56.8 [g] | Yes (by *L. brevis*) | [71] |

**Table 1.** *Cont.*

| Microorganism | Feedstock | Pretreatment [a] | Saccharification | Fermentation Conditions | Lactic Acid | | | | Xylose Utilization | Reference |
|---|---|---|---|---|---|---|---|---|---|---|
| | | | | | Yield | Titer (g/L) | Productivity (g/L·h) | Optical Purity (%) | | |
| *Acremonium thermophilus* ATCC 24622 + *Rhizopus* sp. MK-96-1196 | Corncob | Milled only | Cellulase produced on-site by *A. thermophilus* | Batch culture at 30 °C, initial pH 4.5, aeration rate of 2 vvm and 10% solids loading | N. R. | 24 | N.R. | N.R. | Yes (by *Rhizopus* sp. MK-96-1196) | [30] |
| *Rhizopus oryzae* NLX-M-1 | Corncob | NaOH at 85–90 °C for 1 h, and neutralized H$_2$SO$_4$ solution | Cellic ® CTec2 at 60 mg protein/g DM | Batch culture at 40 °C, pH > 6.0 and aeration rate of 1 vvm and 10% (*w/v*) solids loading | 0.60 [b] | 60.3 | 1.0 | 100 [g] | Yes | [74] |
| *Trichoderma koningii* | Corn straw | Milled and dipped in an ammonia liquor (8%) | Cellulase produced by *T. koningii* in a citric acid–sodium citrate buffer | Batch culture | 0.204 [c] | 20.4 | 0.283 | N.R. | Yes | [75] |

[a] A washing step with water was performed after pretreatment in every experiment except for those that include a biodetoxification step or state it otherwise. [b] g LA/g loaded biomass. [c] g LA/g cellulose. [d] % with respect to maximum theoretical yield from cellulose. [e] Average from the six fermentation rounds. [f] Based on glucose as the only carbon source. [g] %L-LA. [h] %D-LA. DM; dry matter FPU; filter paper units. N. R.; Not reported.

Moreover, the saccharification of corn straw can be achieved with *Trichoderma koningii*, a saprotroph fungus often used as a biopesticide [76] (Tripathi et al., 2010). It can produce cellulases in anaerobic conditions. These enzymes can degrade the corn straw to glucose, cellobiose and xylose. For the SSF process, the manipulated parameters were the solid to liquid ratio, fermentation time, size of the inoculum and pH [75].

Genetically Modified Strains

Genetic engineering has also been explored in strains used in SSF of corn crop residues (Figure 4 and Table 2). The already modified strain of *L. plantarum* NCIMB 8826 Δ*IdhL1* was designed with the purpose of producing optically pure D-LA, by the deletion of its L-lactic acid dehydrogenase gene [77]. This strain was further modified by Y. Zhang et al. (2016) with a recombinant xylose assimilation plasmid. The new *L. plantarum* NCIMB 8826 Δ*IdhL1* -pLEM-*xylAB* strain was able to simultaneously ferment both glucose and xylose into D-LA, with a high yield of 0.77 g/g of pretreated corn stover.

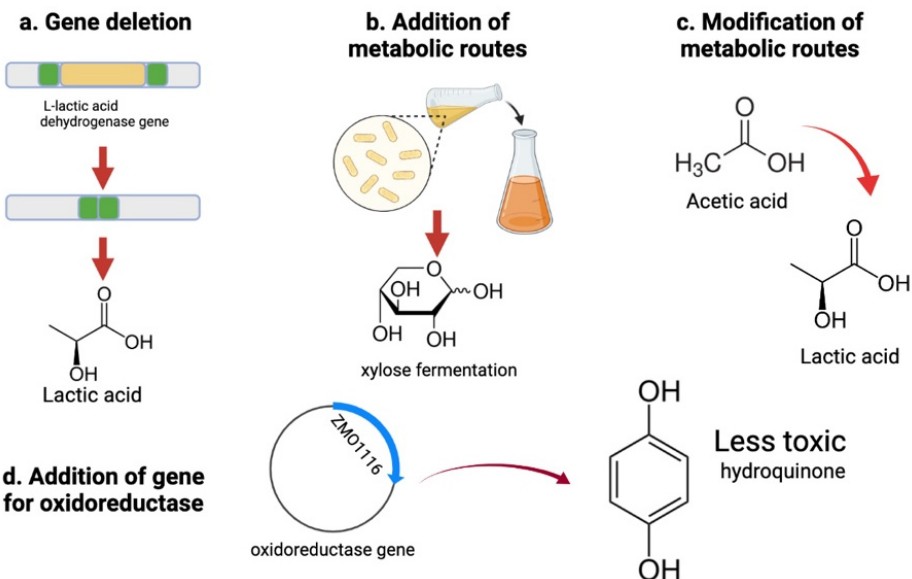

**Figure 4.** Summary of the genetic engineering approaches that have been explored. (**a**) Gene deletion is focused on the production of optically pure LA. *P. acidilactici* ZP26 and *P. acidilactici* TY112 are a product of this modification. (**b**) The insertion of xylose uptake genes resulted in the strains *P. acidilactici* ZY15, *P. acidilactici* ZY271 and *L. plantarum* NCIMB 8826 Δ*IdhL1* -pLEM-*xylAB*. (**c**) *P. acidilactici* ZY15 and *P. acidilactici* ZY271 were additionally modified to decrease acetic acid production. (**d**) *P. acidilactici* ZY15 was transformed to express an oxidoreductase enzyme, which catalyzes the reduction of the harmful *p*-benzoquinone into a safer hydroquinone.

The performance shown by the *P. acidilactici* DQ2 strain in [24] attracted interest for further experimentation using genetic engineering techniques. Two genetically modified strains were created from *P. acidilactici* DQ2 by knocking out the L-lactic acid and D-lactic acid dehydrogenase genes, respectively. *P. acidilactici* ZP26 was able to produce D-LA with 99.32% optical purity, while *P. acidilactici* TY112 produced L-LA with optical purity of 99.89% [34].

**Table 2.** Methodology and results from SSF experiments using genetically-engineered microorganisms.

| Microorganism | Feedstock | Pretreatment | Enzymes | Fermentation Conditions | Lactic Acid | | | | Xylose Utilization | Reference |
|---|---|---|---|---|---|---|---|---|---|---|
| | | | | | Yield | Titer (g/L) | Productivity (g/L·h) | Optical Purity (%) | | |
| *Lactobacillus plantarum* NCIMB 8826 *δldhL1* -pLEM-*xylAB* | Corn stover | NaOH pretreatment at 121 °C for 30 min and water washed | Cellic® CTec2 at 5.6 FPU/g DM | Fed-batch culture at 37 °C and pH 5 | 0.77 [a] | 61.4 | 0.32 | >99 [e, g] | Yes [g] | [60] |
| *Pediococcus acidilactici* TY112 | Corn stover | Dry dilute sulphuric acid pretreatment and biodetoxification | Cellulase Youtell #6 at 15 FPU/g DM Saccharification 6 h prior to inoculation | Batch culture at 45 °C, pH 5.5 and 30% (*w/w*) solids loading | 71.5 [b] | 104.5 | 1.45 | N. R. | No | [23] |
| *Pediococcus acidilactici* ZY271 | Ensiled corn stover | Dry dilute sulphuric acid pretreatment, biodetoxification and partial saccharification. | Cellic® CTec2 at 5 mg protein/g DM. Saccharification 6 h prior to inoculation | Batch culture at 42 °C, pH 5.5 and 30% (*w/w*) solids loading | N. R. | 139.0 | 1.93 [f] | 99.7 [d] | Yes [g] | [33] |
| *Pediococcus acidilactici* TY112 | Corn stover | Dry dilute sulphuric acid pretreatment and biodetoxification | Cellulase Youtell #6 at 15 FPU/g DM. Saccharification 6 h prior to inoculation | Batch culture at 45 °C, pH 5.5 and 25% (*w/w*) solids loading | 65 [b] | 77.66 | 1.06 | 99.89 [d, g] | No | [34] |
| *Pediococcus acidilactici* ZP26 | | | | | 58 [b] | 76.76 | 1.02 | 99.3 [e, g] | No | |
| *Pediococcus acidilactici* ZY15 | Corn stover | Dry dilute sulphuric acid pretreatment and biodetoxification | Cellic® CTec2 at 10 mg protein/g cellulose. Saccharification 6 h prior to inoculation | Batch culture at 42 °C, pH 5.5 and 25% (*w/w*) solids loading | 64.7 [c] | 97.3 | 1.01 | 99.2 [e, g] | Yes [g] | [35] |
| *Pediococcus acidilactici* ZY15-*δackA2::ZMO1116* | Corn stover | Dry dilute sulphuric acid pretreatment and biodetoxification | Cellic® CTec2 at 10 mg protein/g cellulose. Saccharification 6 h prior to inoculation | Batch culture at 42 °C, pH 5.5 and 30% (*w/w*) solids loading | N.R. | 123.8 | N.R. | N.R. | Yes [g] | [61] |

[a] g LA/g loaded biomass. [b] % with respect to maximum theoretical yield from cellulose. [c] % with respect to maximum theoretical yield from cellulose and xylose. [d] %L-LA. [e] %D-LA. [f] Calculated given the data from the article. [g] Derived from genetic modification. DM: dry matter. FPU: filter paper units. N. R.: Not reported.

Although at the time *P. acidilactici* TY112 could not ferment hemicellulose, the strain achieved an industrially attractive L-LA titer (104.5 g/L) [23]. More recently, heterologous genes encoding xylose assimilation enzymes were introduced into *P. acidilactici* ZP26 [35] and TY112 [78]. In these experiments, the metabolic pathway responsible for acetic acid production was re-directed to the LA producing PPP. The new strains were termed *P. acidilactici* ZY15 and *P. acidilactici* ZY271, respectively. *P. acidilactici* ZY15 showed a xylose conversion yield of pretreated corn stover of 92.6% and a final acetic acid titer of 0.50 g/L. On the other hand, *P. acidilactici* ZY271 was first tested in SSF of wheat straw, and reached a xylose conversion of 94.9%, with no carbon catabolite repression. The final acetic acid titer was less than 1.0 g/L. When using corn stover, in a solid-state fermentation followed by SSF, *P. acidilactici* ZY271 reached the highest LA titer out of the reviewed literature (139.2 g/L) [33]. Later, *P. acidilactici* ZY15 was transformed with a plasmid expressing the oxidoreductase gene *ZMO1116*, from *Zymomonas mobilis.* The enzyme converts *p*-benzoquinone, a toxic inhibitor originated in acidic pretreatment of lignocellulose, into less toxic hydroquinone. After SSF at identical conditions, a remarkable 21.4% increase in D-LA titer was observed when compared to its parental strain [61].

## 3. Challenges of Scalability and Trends

### 3.1. Scientific and Engineering Challenges in Bioconversion

In general, the pretreatment and saccharification steps of LB are recognized as the main bottlenecks of industrial scale biorefineries. The negative impact of these operations to the environmental and to the economic feasibility of the complete process tend to increase exponentially as the production is scaled up [79]. Specifically, the experiments reviewed here agreed on two main scaling-up limitations: wastewater generation (especially during pretreatment) and enzymatic efficiency and cost.

3.1.1. Wastewater Generation

The first challenge has to do with the amount of water required by the pretreatment step alone. Moreover, most chemical and physicochemical methods of pretreatment are followed by intensive washing operations. This step ensures the removal of inhibitors of biological activity, and gets the biomass back to a biologically operational pH value. In LA production, 11, 13 and 18 kg of water per kg of dry corn stover were required after steam explosion [80], sodium hydroxide [59] and aqueous ammonia pretreatments [40], respectively (accounting for pretreatment water and washing water). This consumption not only hinders the sustainability of this revalorization scheme, but it also increases the cost of wastewater treatment and there is washout of solubilized fermentable sugars. Different strategies are trying to address removal or inactivation of inhibitory species by less resource-demanding operations. To this end, biodetoxification, alkaline–oxidative post-treatment and genetic engineering are the current strategies for resource optimization of corn residue revalorization. The genetically modified *P. acidilactici* ZY15 has already been mentioned in the previous section.

In the diluted sulphuric acid pretreatment of corn crop residues, the introduction of a biodetoxification step with *Amorphotheca resinae*, a kerosene fungus, is a current trend with interesting outcomes (Table 1). *A. resinae* ZN1 utilizes lignin degradation products as a substrate for microbial growth [81]. The detoxification culture consists of the inoculation of *A. resinae* ZN1 spores in pretreated LB. This strain has demonstrated removal of up to 90% of furfural and 5-hydroxymethylfurfural (HMF) from diluted acid-pretreated corn stover [24]. Biodetoxification enables the incorporation of the whole pretreatment slurry to the fermentation step, thus avoiding the generation of up to 18 kg of wastewater, per kg of dry corn stover [23].

Alkaline–oxidative post-treatment has also been proposed as a means to reduce wastewater generation. The initial substrate is exposed to a $NH_3$ pre-extraction, which aids in the degradation of lignin. A subsequent $H_2O_2$ post-treatment further affects the lignin structure, but also contributes to oxidizing the phenolic compounds generated in the first

stage [82]. A fed-batch SSF experiment was performed to compare the effect on LA titer of two pretreatment methods. Corncob pretreated by a standard three-hours long NaOH-pretreatment protocol, including a washing step, resulted in a LA titer of 122.99 g/L. On the other hand, 84.46 g/L of LA were obtained by using corncob pretreated for eight days with a $NH_3$-$H_2O_2$ protocol [32]; this is one of the highest LA titers obtained from unwashed pretreated LB. Although the current technology is time consuming, more experimentation on this subject could result in a sustainable and feasible pretreatment.

In addition to searching for more efficient detoxification methods for pretreated biomass, wastewater usage is also being tackled by reducing the source of inhibitory compounds. As referred to in the pretreatment section, water soluble hexoses might react with the acid catalyst used for pretreatment, to mainly yield HMF. Thus, a new strategy is based on the bioconversion of water-soluble carbohydrates in a solid-state fermentation of untreated corn stover [33]. This action results in the generation of a LA fraction, which is not significantly affected by subsequent pretreatment and biodetoxification operations, and therefore it is added to the SSF titer. In this experiment, HMF generation was reduced by almost 60%, which in turn allowed for a 33.33% time reduction in the biodetoxification step and a higher L-LA titer (Table 2), which translates into lower purification costs. The solid-state fermentation took place by ensilage of the corn stover, consequently having a low impact in overall costs.

The efforts put into waste reduction in bioprocesses are important, but they cannot reach a level of zero-waste generation, especially when implemented at industrial-scale production. In this sense, a complementing approach, as pointed out in [83], needs to involve the development of a framework for the recovery of potentially valuable compounds and/or energy from wastewater. This could create additional revenue streams for industrial bioprocesses in the context of CE. A suitable solution for resources recovery from SSF wastewater of corn crop residues should be customized, based on the chemical profile of the waste stream, given in turn by the selection of the processing method.

### 3.1.2. Enzyme Efficiency and Cost

Cellulases are the main group of enzymes used in biorefinery of LB, but they have other applications in the food, textile, and paper industries. They are produced by different manufacturers around the world, being Novozymes (Denmark), Genencor-Danisco (USA), and DuPont (USA) the largest ones [84]. Overall, there has been gradual improvement on the bioconversion yield of different formulations depending on the application [85], but there is still room for improvement before reaching cost-effective applications for biorefinery of LB at commercial scale.

The enzyme loadings required to degrade the recalcitrant lignocellulosic matrix depend on the formulation and specific activity of the cocktail used. On average, 15–30 FPU/g corn stover are required to generate a stable flow of reducing sugars to be simultaneously converted to LA. The cost of enzymatic cocktails range between 0.177 and 0.724 USD/kg of LA. These costs position the saccharification as the second most expensive operation in terms of materials [23]. Therefore, these costs must be reduced for saccharification scaling-up.

One of the alternatives to address costs reduction is recycling of the enzymes. After one round of saccharification, a significant amount of enzyme remains active but unproductively bound to specific sites on lignin [86] and, to a lower degree, to cellulose [87,88]. Cellulases like CBHI possess a carbohydrate-binding module (CBM), that binds to the substrate as a first step of the hydrolysis mechanism. Non-productive binding with lignocellulose is mediated by the CBM. In the case of lignin, hydrogen bonds, electrostatic and hydrophobic interactions with the CBM inhibit saccharification [89]. This residual catalytic activity has been harnessed for subsequent saccharification rounds in ethanol production [90], but there are only a few implementation studies of recycling strategies in LA fermentation [68].

Additional accessory enzymes might be included in the reaction to act in synergy with the conventional cellulase cocktail. These can be added as protein isolates, or they can be secreted by degrading fungi. For example, auxiliary activity family 9 enzyme (AA9), a lytic polysaccharide monooxygenase enzyme (LPMO) from *Thermoascus aurantiacus*, improves cellulose accessibility and contributes positively to the release of cellulase from lignin [91]. Taking advantage from this, a commercial cellulase mixture was supplemented with AA9 and xylanases for the saccharification of steam-pretreated corn stover. This strategy was useful to reduce the load of commercial cellulase required for reactions at high substrate concentrations. The hydrolysis yield was increased by 20%, with respect to the unsupplemented reaction, while maintaining the total protein load constant between both experiments [92].

In a more recent experiment focused on LA production by *B. coagulans* LA204, a mixture of hydrolytic enzymes, produced by *Trichoderma viride* R16 in a corncob substrate, was used in the SSF of pretreated corncob. Among these on-site produced enzymes was dioxygenase, which degrades phenolic inhibitors generated in alkaline pretreatment. The fed-batch fermentation resulted in an increase of 24% in LA titer compared to a commercial enzymatic mixture [57].

Other research lines try to improve the catalytic activity of cellulases through directed evolution and/or computer-aided protein engineering, to reduce enzyme loadings. For instance, in [93], the activity of endoglucanases from *T. reseei* (Cel12A) was improved by a DNA-shuffling approach. In other studies, non-productive adsorption of cellobiohydrolases from *T. reseei* (*Tr*Cel7A) to lignin was reduced, by including negatively charged residues to their CBM [94]. Moreover, cellulase activity is linked to the binding affinity to their substrate. This information represents new opportunities for in silico design of cellulases [95]. More promising protein engineering approaches are reviewed in [96]. This provides evidence that reducing saccharification-related costs is one of the main research areas in the development of sustainable bioconversion platforms.

### 3.2. Challenges Associated with SSF as Bioconversion Platform

SSF and its variants emerged as bioconversion alternatives that allow higher solids loadings when compared to SHF, while avoiding cellulase inhibition by glucose or cellobiose [20]. Nonetheless, SSF also presents drawbacks that are inherent to the integration of saccharification and fermentation in a single-vessel operation.

As previously pointed out, the main disadvantage of SSF is that optimal conditions for lignocellulose saccharification and LA fermentation are often different. Specifically, cellulases and other degrading enzymes show their best catalytic performance at temperatures between 45 and 50 °C [46]. Thus, a compromise is often made to favor one operation over the other in SSF. This trade-off does not represent an issue in SHF, as enzymatic hydrolysis and fermentation conditions are optimized independently [97].

Another drawback that is adverted to about SSF is the competition for chemical resources between both bioconversion steps. An example of this is when a LPMO, such as the one described in the previous section, is part of the saccharification cocktail. LPMOs need molecular oxygen to cleave cellulose. In a SSF set up, with an aerobic strain, oxygen uptake favors fermentation, thus creating a non-optimal environment for LPMO action [98]. This effect where SHF outperforms SSF in the presence of LPMO has been observed in alcoholic [99] and LA fermentation [100].

### 4. Feasibility Analysis of SSF of Corn Crop Residues for LA Production
#### 4.1. Technology Readiness Assessment

Besides closing current linear production practices, biorefining also attempts to lower manufacturing costs by utilizing cheap resources. Consequently, the previously discussed challenges and some others arise. Therefore, SSF of corn crop residues must meet certain milestones before becoming a platform for commercial production of LA. In this section, a technology readiness level-based assessment is used to put progress of this approach

into perspective. The technology readiness level (TRL) scale goes from 1 to 9, and it is a worldwide communication tool that depicts the level of maturity of a certain technology. In the field of biorefining, this methodology has been used before to assess the implementation of different resource recovery projects [5].

TRL can be used to evaluate and compare technologies from different fields. In this sense, the TRL-based assessment presented in Table 3 is based on the chemical industry-specific criteria and indicators proposed in [101]. The framework is divided into four project categories: description, general project criteria, engineering criteria and capacity. Each category is assigned its own TRL scale paired with indicators for each level. These are briefly described and put in context using the technology under investigation. The assessment is limited to upstream operations, and it sets out to answer the question: *to what degree is LA SSF of corn crop residues ready for commercial application?* In the end, the general TRL is equal to the lowest TRL out of the four criteria.

The description category outlines the main completed activities and achievements within a project. The indicators associated with the TRL scale range from potential applications of basic research at TRL 1, full-scale simulations at TRL 5, and full-scale plant audited at TRL 9. In this case, SSF of corn crop residues has been studied in a laboratory environment (Tables 1 and 2), and full-scale simulations using bench-scale information [102], with just a few patented methodologies [103]. These milestones are appropriate to TRL 5 in the description category.

The general project criteria is subdivided into tangible work results, workplace and product specifications. The indicators of this category keep track of the project's progress in terms of the reproducible experimental work and results, the infrastructure, and the technical specifications of the product. In this sense, LA is a well-characterized product that is sold in an 88% solution. However, the literature review reveals an absence of SSF scaling-up studies, with 50 L being the maximum fermentation volume reported [24]. The recent publications seem to be rather directed to testing of different genetically modified microorganisms, meaning that alternative process variables are still under evaluation. As a result of the underdeveloped and small-capacity process, the general project criteria is set to TRL 5.

The TRL in the engineering category is defined by the degree of knowledge about the metabolic reactions involved. Each level also states milestones on process design such as unit operations, energy flows, equipment specifications, and flow diagrams. In this case, the different microbial metabolic pathways that transform simple carbohydrates into LA are well understood (Figure 3). However, less attention has been directed towards customized pretreatment and enzymatic hydrolysis of corn crop residues. Regarding the process, different combinations of pretreatment methods, enzymatic cocktails and microorganisms are still being tested, resulting in high conversion yield [32], high LA titer [33] and optical purity [34]. Detailed kinetic studies for most SSF systems are available and the unit operations are well-defined, together with ranges of operating conditions. With these characteristics, engineering criteria is positioned at TRL 4.

Lastly, capacity TRL is defined as a fraction of full-scale production and the type of product. LA production in a single commercial plant is between 50 and 200 kton a year, which represents a scale-up factor of 100,000 compared to the highest tested capacity of SSF of corn crop residues [24]. This scale-up factor is paired to TRL 3 for commodity chemicals such as LA.

To find the overall TRL rating, the authors of [101] suggest to take the "the weakest link in the chain", which in this case corresponds to TRL 3 for the capacity criterion. The leading LA manufacturers are NatureWorks LLC and Corbion, and they base their process in 1G-feedstocks. These companies are also investigating 2G- and 3G-LA technologies, but large-scale adoption of these approaches is not reflected yet in publicly available sources.

**Table 3.** Assessment of technology readiness level for production of LA.

| Description | Maturity | TRL | | | | Reference |
|---|---|---|---|---|---|---|
| | | Description | General Project | Engineering | Capacity | |
| SS(C)F of corn crop residues for LA manufacture | Batch culture at high solids loadings Low wastewater generation High optical purity, titer, and bioconversion yield Mini-plant scale fermentation Patents | 5 | 5 | 4 | 3 | [24,33,102,103] |
| Homofermentation of 1G feedstock | Commercial production by NatureWorks LLC, Galactic, Corbion, etc. | 9 | 9 | 9 | 9 | |

*4.2. Review on Techno-Economic Evaluation*

Several (techno-economic analysis) TEAs have addressed manufacturing of 2-G LA using corn crop residues and different process configurations. The findings and results vary within studies, but the main conclusion remains: large-scale SSF of corn crop residues is likely to be financially viable. These results come from simulations based on extrapolated bench-scale results and assumptions, and they consider mature technologies and acceptable return on investments (ROI) of at least 8%. The market price of LA ranges between 3.0 and 4.0 $USD/kg [13]. LA is often traded as an 88% solution, which is considered for most full-scale simulations. Although downstream processes are out of the scope of this review, these are necessarily considered in every TEA.

In the TEA in [22], 2000 metric tons/day of heterogeneous LB was set as feedstock of a conventional LA bioconversion route (dilute sulphuric acid pretreatment and SHF). The MPSP of 2G-LA was on average 10% lower than the market price of the biochemical. However, the environmental impact of the biorefining process (as measured by fossil energy consumption and 100-year global warming potential categories) could be higher or close to fossil-derived LA. In this case the conversion process accounted for an average of 30% of the material costs. The TEA study in [23] compared SSF of corn stover using cellulases produced on-site and commercial cocktails. In the latter scenario, the MPSP of L-LA was increased up to 108%; still, according to this simulation, the SSF using commercial enzymes results in low MPSP ($0.523–$1.166/kg). Moreover, in a scenario where water-soluble carbohydrates were pre-fermented, and a xylose fermenting strain was used for SSCF of corn stover, the MPSP achieved the lowest reported value of $0.459/kg [33].

*4.3. SWOT Analysis*

The SWOT analysis on Figure 5 summarizes some of the topics addressed in this article that define the competitiveness of SSF of corn crop residues for LA production. Biorefining is generally assumed to be a more sustainable manufacturing platform than chemical synthesis, because it is based on cheap substrates and microbial metabolism. The present work reveals that the sustainability picture goes beyond substrate source and conversion route. For instance, there are environmentally harmful waste streams generated along the pretreatment [61], the bioconversion [13], and the downstream operations [7]. Moreover, after LA fermentation there are significant amounts of residual substrate and materials (lignin, biocatalysts, etc.) in the fermentation broth, thus resulting in their loss.

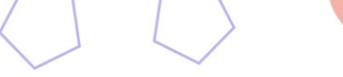

**Strengths**

- In line with CE principles and SDG.
- Improved performance compared to SHF.
- More feedstock can be processed per fermentation.
- Lowered investment and maintenance costs, compared to SHF.
- Feedback inhibition is absent or limited.
- Renewable feedstock.
- Generation of high titer and optically pure LA, with low fermentation by-products.

**Weaknesses**

- Polluting waste streams from bioconversion and downstreaming.
- High generation of wastewater.
- Technology is not directly transferable to different raw materials.
- The saccharification rate holds back the fermentation rate.
- Residual substrate and active enzyme in fermentation broth.
- Catabolic repression.

**Opportunities**

- Creation of additional revenue streams by managing waste.
- Platform for $CO_2$ biocapture.
- Platform for replacement of petrochemicals.
- Further development for biomanufacturing of specialty chemicals.
- Potential cost reduction with open fermentation systems.
- Corn is the 4th largest crop worldwide.
- Ongoing research to address scaling-up issues.

**Threats**

- CBP could outperform SSF at some point.
- Large production required to be cost-effective.
- Bioconversion of 1G- and 3G- feedstock is cheaper.
- Difficult to scale-up.
- Lack of, or unclear regulations.
- Cellulase cocktails are costly.

**Figure 5.** SWOT analysis of the bioconversion of corn crop residues to LA through SSF.

There is not a single LA conversion framework capable of checking all the desired boxes. Chemical synthesis, SHF, SSF and CBP have each their advantages and disadvantages. However, recent research has focused on optimization of SSF and CBP, which is one of the reasons a review of the SSF technology was deemed necessary.

## 5. Conclusions

The reviewed articles suggest that saccharification is the main limiting step in the bioconversion of corn crop residues to 2G-LA. Moreover, wastewater generation and cost of enzymes have limited scalability and maturity for commercial-scale implementation of this technology (as well as other biorefining setups), emphasizing pretreatment and saccharification steps as the key targets for future research. Additionally, SSF presents drawbacks that are inherent to the combination of the bioconversion steps. These are related to the optimization of reaction conditions and competition for chemical resources. On the other hand, fermentation optimization is narrowed down to the use of *B. coagulans* and *P. acidilactici* species, the latter being subject to successful genetic modifications that improve performance and purity of the final product. Therefore, SSF of corn crop residues is a potential manufacturing platform for sustainable LA production.

**Author Contributions:** Conceptualization, A.M.-B. and E.M.M.-M.; methodology, H.M.N.I.; software, A.M.-B.; investigation, S.S.M.; writing—original draft preparation, A.M.-B.; writing—review and editing, J.E.S.-H. and E.M.M.-M.; visualization, L.M.R.-J.; supervision, E.M.M.-M. project administration and funding acquisition R.P.-S. and E.M.M.-M.; All authors have read and agreed to the published version of the manuscript.

**Funding:** This research received no external funding.

**Acknowledgments:** This work was partially supported by the Consejo Nacional de Ciencia y Tecnología (CONACyT) and Tecnologico de Monterrey through the scholarship awarded to the first author (Alonso Malacara-Becerra, CVU: 894370). CONACyT is thankfully acknowledged for partially supporting this work under the Sistema Nacional de Investigadores (SNI) program awarded to Juan Eduardo Sosa-Hernández (CVU: 375202), Elda M. Melchor-Martínez (CVU: 230784), Hafiz M.N. Iqbal (CVU: 735340), and Roberto Parra-Saldívar (CVU: 35753). The authors acknowledge the support of the participant institutions for gaining access to scientific journal databases. The graphical abstract and figures were created with BioRender.com.

**Conflicts of Interest:** The authors declare no conflict of interest.

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
