# Peer review of "Bioconversion of Corn Crop Residues: Lactic Acid Production through Simultaneous Saccharification and Fermentation"

_sustainability, doi:10.3390/su141911799_

Round 1

Reviewer 1 Report

Overall, this review is beneficial to the area of Bioconversion/biorefinery. The review on the microorganisms part is quite extensive. However, some parts of the reviews are too shallow, and not much discussion or critical review could be obtained.

From the article title, it will cover agriculture residue. However, the content focuses on corn residue (even not comprehensive).

After introducing 3 methods of LB biorefinery in Fig 2, There is no discussion on why SSF has been chosen to be reviewed in this paper. Only one sentence on the advantage of the SSF compared to SHF based on ref patent [19]. Some limitations and advantages of one pot system also could be highlighted.

It is good to have some sort of conclusion for each section rather than an open statement. For instance, in section 2.1.1: Basically, only 2 examples have been stated, ref [27] and [24], even though others have conducted so many studies in this area. Moreover, the last sentence stated the upscaling problem and did not conclude the chemical pretreatment.

Is there any guideline, characteristics, or procedure to have in order to select SSF compared to conventional SHF? It should be described clearly in the manuscript.

There is also no conclusion for one of the most important reviews in section 2.2.1. If it is for corn crop residues, what is the best enzyme and production, and how does it relates to the simultaneous conversion during fermentation in SSF? The topic has been discussed separately as enzyme hydrolysis and fermentation; however, some interaction points and conditions need to be discussed or reviewed.

Author Response

Dear reviewer,

We wish to thank your thoughtful feedback on this paper. We have addressed your suggestions as follows in the paragraphs below:

Reviewer 1 comments

Overall, this review is beneficial to the area of Bioconversion/biorefinery. The review on the microorganisms part is quite extensive. However, some parts of the reviews are too shallow, and not much discussion or critical review could be obtained.

Author's response: Thanks to the sum of your assertive feedback, it was possible to add deeper discussions to most sections of the article. Specifically in lines 83-90, 136-149, 178-180, 225-233, 294 to 298-308, 314-325, 491-496, and 605-621. We believe that these changes resulted in a more critical and comprehensive review.

From the article title, it will cover agriculture residue. However, the content focuses on corn residue (even not comprehensive).

Author's response: Thank you for your valuable comment. Corn crop residues are one of the main feedstocks under investigation in biorefining. Therefore, we believe that the material covered in this article will provide the essential starting point for new research. Thanks to your observation we decided to change the title of the article, to avoid ambiguity regarding its content.

After introducing 3 methods of LB biorefinery in Fig 2, There is no discussion on why SSF has been chosen to be reviewed in this paper. Only one sentence on the advantage of the SSF compared to SHF based on ref patent [19]. Some limitations and advantages of one pot system also could be highlighted.

Author's response: Thank you for assertive suggestion. In the introduction, lines 83-102 were modified to include more advantages of SSF compared to SHF. This also served to justify the reviewing of this bioconversion framework. The reader is also referred to section 2.2 (Please see lines 231-251), where more advantages and limitations of the technology are addressed. This information was kept out of the introduction to avoid extending the section further.

It is good to have some sort of conclusion for each section rather than an open statement. For instance, in section 2.1.1: Basically, only 2 examples have been stated, ref [27] and [24], even though others have conducted so many studies in this area. Moreover, the last sentence stated the upscaling problem and did not conclude the chemical pretreatment.

Author's response: Thank you for your observation. More examples and sources were added to section 2.1.1. Now the section ends with a practical conclusion to physical pretreatment. Please see lines 142-149. Also, section 2.1.2 now concludes chemical pretreatment by referring to the most popular methods. Please see lines 178-180. The upscaling issues are mentioned in this section, but the reader is referred to section 3.1, where they are addressed with more detail. Section 2.2.1 was also modified in lines 314 through 325 to have a proper conclusion to the section instead of an open statement.

Is there any guideline, characteristics, or procedure to have in order to select SSF compared to conventional SHF? It should be described clearly in the manuscript.

Author's response: Thank you for pointing this out. We agree with this comment. However, there are no guidelines to favor SSF instead of SHF, because the former tends to outperform the latter. Therefore, the decision to select one over the other should be based on strain availability. This is stated in lines 243-246.

There is also no conclusion for one of the most important reviews in section 2.2.1. If it is for corn crop residues, what is the best enzyme and production, and how does it relates to the simultaneous conversion during fermentation in SSF? The topic has been discussed separately as enzyme hydrolysis and fermentation; however, some interaction points and conditions need to be discussed or reviewed.

Author's response: Thank you for your good suggestion. To address this comment, new information was added in lines 294-295, 298-308, and 314-325. There, Cellic ® CTec2 is defined as the main enzymatic cocktail used for saccharification of corn crop residues, due to the higher saccharification yield when compared to other formulations. Also, it is stated that there is a lack of studies on the set of these conditions that synergically benefit saccharification and fermentation. Instead, fermentation conditions are favored most of the time.

Reviewer 2 Report

The use of agricultural waste as an input for a biorefinery process for the production of lactic acid has been extensively studied in recent years. However, the optimization of the enzymatic process of corn stover valorization is a topic to be explored.

In fact, the authors have as main purpose of this review the study of the recent biorefinery optimizations for the production of LA using corn wastes.

The introduction is exhaustive, as well as the description of the pre-treatments on biomass and fermentations by LAB. The references used are quite recent.

However, I found the section on production costs and process scalability poorer than the introductory sections. In my opinion, this needs to be improved before accepting the manuscript because this is what makes this review work original. 
Furthermore, I would suggest the authors to add techno-economic opinions on lactic acid production processes using agricultural waste. In fact, many TEA papers have been published regarding the production of lactid acid from agricultural and corn waste. This could also help to specify the costs of enzymes used (which were not well defined in paragraph 3.2).

In the section 4, i could not find figure 5 in the manuscript, why this is missing? However, it would be useful for the authors to spend a few more words on the results of this SWOT analysis, then I did not fully understand how the technology readiness level (TRL) was conducted, it would be better to specify these details.

Author Response

Dear reviewer,

We wish to thank your thoughtful feedback on this paper. We have addressed your suggestions as follows in the paragraphs below:

Reviewer 2 comments

The use of agricultural waste as an input for a biorefinery process for the production of lactic acid has been extensively studied in recent years. However, the optimization of the enzymatic process of corn stover valorization is a topic to be explored.In fact, the authors have as main purpose of this review the study of the recent biorefinery optimizations for the production of LA using corn wastes.

The introduction is exhaustive, as well as the description of the pre-treatments on biomass and fermentations by LAB. The references used are quite recent. However, I found the section on production costs and process scalability poorer than the introductory sections. In my opinion, this needs to be improved before accepting the manuscript because this is what makes this review work original.

Author's response: Thank you for your valuable comment. The introduction was reconstructed. The section on scalability limitations was improved by adding information regarding production costs in lines 560-565.

Furthermore, I would suggest the authors to add techno-economic opinions on lactic acid production processes using agricultural waste. In fact, many TEA papers have been published regarding the production of lactid acid from agricultural and corn waste. This could also help to specify the costs of enzymes used (which were not well defined in paragraph 3.2).

Author's response: Thank you for noticing it. Section 4 was renamed and divided into 3 subsections, one of which addresses the TEA of the bioconversion framework under study (section 4.2) in lines 685-705. This new addition highlights the financial feasibility of the technology.

In the section 4, i could not find figure 5 in the manuscript, why this is missing? However, it would be useful for the authors to spend a few more words on the results of this SWOT analysis, then I did not fully understand how the technology readiness level (TRL) was conducted, it would be better to specify these details.

Author's response: Thank you for the observation on Figure 5. It was added to the MS. Moreover, the SWOT analysis is now described in 2 new paragraphs. Please see lines 707-719. Also, the technology readiness assessment is now described with more detail in section 4.1. Please see lines 624-679.

Reviewer 3 Report

Please see the attached document titled 'Comments_to_authors.pdf'.

Author Response

Dear reviewer,

We wish to thank your thoughtful feedback on this paper. We have addressed your suggestions as follows in the paragraphs below:

Reviewer 3 comments

The authors do a good job of providing the reader with enough background to biomass deconstruction and laying out the advantages of simultaneous saccharification and fermentation. However, there are certain areas that could use improvement, to provide a holistic perspective to the readers

There is not a significant discussion of the drawbacks of SSF approaches such as for instance, the trade-off between optimal temperature for enzymatic hydrolysis vs saccharification. The authors need to include a discussion both in section 2.2 (page 6) and eventually in the conclusions as well, to provide the reader with a balanced perspective on the pros and cons of SSF approaches

Author´s response: Thank you for the suggestion. Discussion on section 2.2 was extended a few more lines on the subject of the characteristics of SSF (Please see lines 314-325). However, as part of your following feedback, the new section 3.2. Challenges associated with SSF as bioconversion platform” addresses a deeper discussion on drawbacks of SSF (Please see lines 605-621). Additionally, these were also added to the final remarks of the document.

On page 7, the paragraph that starts with ‘Most of the studies..’, the authors mention that the compositions of enzyme cocktails such as Cellic C.Tec2 have not been evaluated so far. There has been work performed on using proteomics-based approaches for decoding the enzyme cocktail compositions. Authors can read through these references to understand what the ideal composition of cocktails could be, for improved biomass deconstruction. 

Author's response: Thank you for your valuable contribution. Proteomic-based approaches are now discussed in lines 298-308. Also, filamentous fungi are mentioned as the main source of industrially produced cellulases. These new references help elucidate the nature of the hydrolytic enzymes required for an effective saccharification.

  1. In the section on ‘3. Challenges of scalability and trends’ on page 16, the authors list wastewater treatment and enzyme efficiency and cost. This is not an exhaustive list of factors that plague the lignocellulosic biomass deconstruction industry. The authors can divide this section into two sub-sections: 1. Scientific and engineering challenges faced by biomass industry in general (regardless of whether the format is separate hydrolysis and fermentation or simultaneous saccharification and fermentation) 2. Challenges associated with the SSF format vs separate saccharification and fermentation.

 Author's response: Thank you for your contribution to the layout of the manuscript. Section 3 is now divided into 2 separate subsections, where 3.1 addresses the main challenges common in any bioconversion set up (exemplified by the reviewed SSF studies) and subsection 3.2 addresses the main challenges of a SSF bioconversion set up. 

In the sub-section 3.2 on ‘Enzyme efficiency and cost’, two improvements can be made:

  1. there is currently literature on engineering cellobiohydrolase enzymes and endocellulases so as to improve their activity and reduce enzyme loading. This literature should be discussed by the authors to present a hopeful perspective of how the enzyme cost related challenges can be overcome
  2. the authors mention only non-productive binding to lignin but there is evidence to show that non-productive binding to cellulose and hemicellulose especially via the CBM could also be a potential concern

Author's response: Thank you for making these observations. First of all, protein engineering approaches are now discussed in section 3.1.2 (Please see lines 594-603), adding to the list of approaches focused on improving lignocellulose saccharification. Also, it is now pinpointed in lines 569-573 that non-productive binding of cellulases also concerns cellulose and hemicellulose.  

To frame the reader’s perspective on the economic challenges involved, the authors need to provide a better review of the literature on techno-economic analyses in section 4 on page 18. In the current version, there is a minor discussion in the sub-section on enzyme efficiency and cost but that is not exhaustive

Author's response: Thank you for your valuable comment. The new subsection “4.2 Review on techno-economic evaluation” is now part of the manuscript. Here, the literature on the TRL of SSF of corn residues for LA production is presented, to provide the reader with a comprehensive view on the feasibility of SSF.

Round 2

Reviewer 1 Report

All the concerns have been addressed appropriately by the authors.

Reviewer 2 Report

The authors have answered all my questions, the quality of the article has improved and it is ready to be published.

I would just do some English checking of the corrections made.

Reviewer 3 Report

Thanks for addressing the concerns raised